# Investigating barriers & facilitators for the successful implementation of the BP@home initiative in London: Primary care perspectives

Eva Riboli-Sasco[1], Austen El-Osta[1]*, Marie Line El Asmar[1], Manisha Karki[1], Gabriele Kerr[2], Ganesh Sathaymoorthy[2], Azeem Majeed[1]

**1** Self-Care Academic Research Unit (SCARU), Department of Primary Care & Public Health, Imperial College London, London, United Kindgom, **2** NIHR ARC North West London, London, United Kingdom

* a.el-osta@imperial.ac.uk

## Abstract

### Background

The COVID-19 pandemic led to the implementation of a national policy of shielding to safeguard clinically vulnerable patients. To ensure consistent care for high-risk patients with hypertension, NHS England introduced the BP@home initiative to enable patients to self-monitor their blood pressure by providing them with blood pressure monitors. This study aimed to identify barriers and facilitators to the implementation of the initiative based on the experience and perspectives of programme managers and healthcare professionals (HCPs) involved in its implementation in London.

### Methods and findings

We conducted five semi-structured focus groups and one individual interview with a total of 20 healthcare professionals involved at different levels and stages in the BP@home initiative across four of the five London Integrated Care Systems (ICSs). All focus groups and interviews were audio-recorded, transcribed and analysed thematically following the Framework Method. Respondents reported being challenged by the lack of adequate IT, human and financial resources to support the substantial additional workload associated with the programme. These issues resulted in and reinforced the differential engagement capacities of PCNs, practices and patients, thus raising equity concerns among respondents. However respondents also identified several facilitators, including the integration of the eligibility criteria into the electronic health record (EHR), especially when combined with the adoption of practice-specific, pragmatic and opportunistic approaches to the onboarding of patients. Respondents also recommended the provision of blood pressure monitors (BPMs) on prescription, additional funding and training based on needs assessment, the incorporation of BP@home into daily practice and simplification of IT tools, and finally the adoption of a person-centred care approach. Contextualised using the second iteration of the Consolidated Framework for Implementation Research (CFIR), these findings support key evidence-

**Data Availability Statement:** All relevant data are within the paper and its Supporting Information files.

**Funding:** The authors received no specific funding for this work.

**Competing interests:** The authors have declared that no competing interests exist.

based recommendations to help streamline the implementation of the BP@home initiative in London's primary care setting.

## Conclusions

Programs such as BP@Home are likely to become more common in primary care. To successfully support HCPs' aim to care for their hypertensive patients, their implementation must be accompanied by additional financial, human and training resources, as well as supported task-shifting for capacity building. Future studies should explore the perspectives of HCPs based in other parts of the UK as well as patients' experiences with remote monitoring of blood pressure.

## Introduction

Noncommunicable diseases (NCDs) are responsible for 71% of deaths globally, with cardiovascular diseases (CVD) accounting for most deaths [1]. Investing in CVD prevention is integral to achieving or at least progressing toward several Sustainable Development Goals [2]. However, many public health policies were affected by the COVID-19 pandemic, which resulted in the rapid implementation of a national government policy of shielding to protect clinically vulnerable patients [3]. This was accompanied by important changes in the provision of NHS primary care services, including a switch from traditional face-to-face appointments to remote assessments and consultations using the telephone or videocalls [4]. This meant that shielded patients with uncontrolled hypertension could no longer safely access blood pressure (BP) monitoring in person. Loss of follow-up meant that healthcare professionals (HCPs) were unable to provide tailored interventions to control the patient's BP and medication dosage.

Delays of only a few months in medication intensification and BP follow-up are associated with an increased risk of an acute cardiovascular event (CVE) or death, highlighting the importance of timely medical management and follow-up in the treatment of hypertensives [5]. Disruption of only nine months to the delivery of routine care for NHS patients diagnosed with hypertension was estimated to result in almost 12,000 additional acute CVEs, including stroke and heart attack or deaths over a three-year follow-up period [6].

The NHS England-funded BP@home programme [7] was launched in 2020 sought to address this issue. BP@home is part of the larger NHS@home initiative [8], which aims to provide more personalised, convenient, high quality and timely alternatives to face-to-face care by maximising the use of technology to support more people's self-care in the home and community setting. Around 280,000 BP monitors (BPMs) were procured by NHS England to support the BP@home programme. Allocation of BPMs was coordinated by Integrated Care Systems (ICSs) across England, including five across London: North West London (NWL), North East London (NEL), North Central London (NCL), South East London (SEL) and South West London (SWL). BPMs would then be allocated to Primary Care Networks (PCNs), followed by local general practices before finally arriving to eligible patients (**Fig 1**).

The BP@home standard operating procedure (SOP) encouraged each local area and GP practice to decide which patients had the greatest need and should be prioritised for regular home blood pressure monitoring. Recommended criteria for prioritisation included age, BP level, deprivation and pre-existing CVD. A list of search and stratification tools based on the UCLPartners Proactive Care Framework for hypertension was provided to be used within GP systems (SystmOne and EMIS; see **S1 Fig**). This framework is part of several others developed

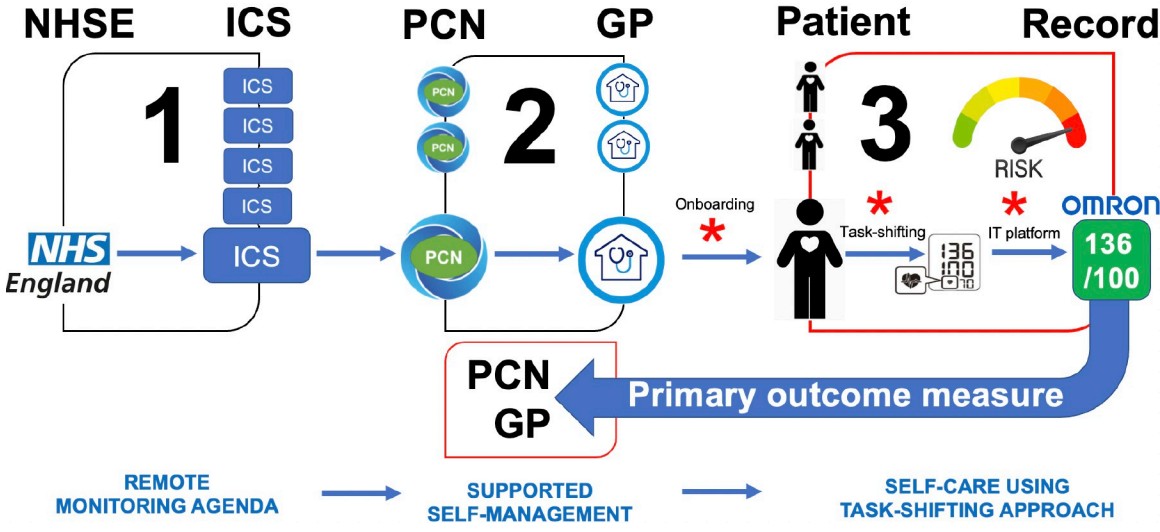

**Fig 1. Schema of BP@home pathway.**

by the UCLPartners to transform the management of high BP, high cholesterol, atrial fibrillation and diabetes in primary care [9]. Once eligible patients were identified, HCPs contacted patients and offered them a BPM for a limited period to maximise their use by other eligible patients. The detailed recommended pathway is presented in **S2 Fig**.

This programme and recommendations were based on evidence showing the benefits of home BP monitoring for improving BP control among users in comparison to standard BP monitoring in the healthcare system [10–12], especially if combined with co-interventions such as the systematic medication titration by doctors, pharmacists, or patients; education; or lifestyle counselling [13]. The titration of antihypertensive medication in primary care using self-monitoring, with or without telemonitoring, also results in significantly lower BP after one year but also that using telemonitoring leads to lower blood pressure after six months [14]. In addition, remote telemonitoring can allow a shift of care from GPs to other members of the multidisciplinary team e.g., healthcare assistants dealing with incoming data and signposting to practice nurses (PNs), pharmacists or GPs as needed. Programmes like BP@home, hold potential beyond the COVID-19 pandemic contexts. However, to obtain these promising results, barriers and drivers to the efficient and widespread implementation of self-monitoring must be identified and addressed.

Recent studies have raised concerns regarding the traceability and reliability of the BP measurements [15], the security of patients' data [16, 17] as well as the costs associated with the training of health care professionals (HCPs) and the implementation and maintenance of the required IT tools [18]. Other sources of concern include the additional workload related to the readings received through telemonitoring [19] and their integration into electronic medical records [20]. Furthermore, IT tools and systems should be more user-friendly, cost-effective, durable and with better safety standards [21]. Socio-economic inequalities and differential abilities have been identified as significant challenges to equal access, use, adoption and impact of eHealth services by patients [22].

The aim of this study was to explore the perspectives of programme managers and healthcare professionals (HCPs) involved at all stages of the BP@home initiative across London. Specifically, we sought to identify extant barriers and facilitators to the streamlined implementation of BP@home from the system level / NHS perspective.

## Methods

### Study design

We used an applied qualitative research study approach using semi-structured interviews and focus groups with 20 HCPs to identify the main barriers and drivers to the successful implementation of the BP@home initiative based on their shared experiences and perspectives.

### Data collection

We used a purposive sampling technique [23]. Potentially eligible participants were contacted through professional contacts via email. We specifically targeted programme managers and HCPs involved in the BP@home initiative in London, either at ICS and PCN levels or at the coalface of primary care and invited respondents to share the study in their networks. Given this recruitment method, it was not possible to know exactly how many potential respondents received the invitation and declined. Potentially eligible participants who expressed an interest or willingness to be interviewed were sent the participant information sheet, which included the theme and aims of the research project, as well as details on the upcoming interviews. Informed consent to be interviewed and audio-recorded was taken first via email and then again verbally just prior to starting the interviews. Respondents were informed they could withdraw at any time and that their participation was voluntary. Of those who responded and agreed to take part, two eventually could not attend any of the focus groups scheduled due to time constraints. However no additional interviews or focus groups were organized as theme saturation had already been achieved. 3 participants were previously known to study authors by virtue of participating in a local clinical research group monthly online meetings.

The research team conducting interviews and focus groups comprised two mixed methods researchers, one male (AEO; MSc, MPA, PhD) and one non-binary (ERS; MA Social Sciences), as well as one female quantitative data analyst (GK; MPH). All members of the study team supported the development of the study protocol, study design, data collection and data analysis, and were experienced in conducting qualitative research. Interviews were conducted remotely via Microsoft Teams between 28 May 2022 and 15 February 2023. AEO, ERS and GK conducted the initial three semi-structured focus groups. AEO and ERS later conducted two additional focus groups and one personal interview. While the individual interview allowed to get into further detail with one respondent, focus groups were prioritised as a data collection tool which can provide in-depth and diverse information from a multitude of respondents in a fast and convenient way [24], even when carried remotely [25]. The aim was also to provide respondents with the opportunity to share their ongoing experience with the implementation of the BP@home program, including practical tips and advice which may support them in their work. Regarding the composition of the focus groups, we aimed to gather together participants we were relatively homogenous in terms of occupation and level of intervention (ICS and PCN levels versus practice and patient facing). This methodological choice aimed to facilitate frank discussions and honest sharing of experiences by limiting power relations [26]. The composition of each focus groups is detailed in **Table 1**.

All focus groups were facilitated by AEO and guided by an interview guide containing a list of questions, some of which targeted more specifically at HCPs working at ICS/PCN level and others at the practice level (**S1 Table**). Each focus group started with a short presentation of the project by the research team, followed by a round table of all participants to share their name and professional designation. Focus groups lasted between 45 and 60 minutes, while the personal interview lasted 45 minutes. All focus groups and personal interviews were audio-recorded, transcribed and thematically analysed. Only two authors, AEO and ERS, had access

**Table 1. Composition of focus groups.**

| Focus Group | Occupation/Role of participants | Number of participants |
|---|---|---|
| 1 | Program leads at ICS & PCN levels | 3 |
| 2 | Practice & Patient level HCPs | 3 |
| 3 | Practice level HCPs and one Program Manager at PCN level | 6 |
| 4 | Practice level HCPs (pharmacist, nurse, GP) | 4 |
| 5 | Program managers (technical & clinical operational leads) | 4 |

to the complete transcriptions of interviews which were saved on Imperial College London's secure online environment that could only be accessed using login. Transcripts were not returned to participants and no feedback was requested from them regarding the analysis. The Consolidated Criteria for Reporting Qualitative Research (COREQ) was used to guide reporting of the study (**S2 Table**). The study received a favourable opinion from Imperial College Research Ethics Committee (ICREC # 22IC7676).

**Data analysis.**   A thematic analysis was conducted on focus groups and personal interview transcripts following the seven stages of the Framework Method [27] to identify codes and construct main themes. Two researchers, AEO and ERS, reviewed the transcripts and agreed on emerging themes, which were later discussed with the research team. Two main classifications were defined prior to the analysis: (1) barriers and challenges, and (2) facilitators and strategies to streamline the implementation of BP@home in the real-world setting. Within these groups, we later identified four sub-categories derived from the data: (i) project management, (ii) logistics (including technical and IT issues), the recruitment and engagement of (iii) PCNs and practices, and (iv) patients. We contextualised our findings using the second iteration of the Consolidated Framework for Implementation Research (CFIR) published in 2022 [28] and proposed evidence-based recommendations to help streamline the implementation of the BP@home initiative in London's primary care setting.

## Results

Following the presentation of the respondents, we first present the barriers and challenges they encountered along the BP@home pathway, and then the facilitators and strategies identified.

### Profile of respondents

We interviewed a total of 20 programme managers and HCPs working in four of the five London Integrated Care Systems (North West London (NWL), North East London (NEL), North Central London (NCL) and South West London (SWL)). All respondents were involved at different levels and stages of the BP@home initiative. They included 15 primary care workers (PCWs) such as GPs, clinical pharmacists, nurses, and seven non-medical respondents with managerial or administrative roles. One GP and one clinical operations lead were employed in the private sector. Apart from two respondents, all other participants (n = 18) were NHS workers. Participant characteristics are summarised in **Table 2**.

### Barriers & challenges

Respondents described several barriers and challenges that they had personally encountered or became familiar with since the launch of the BP@home pathway. As with most large-scale programmes, these barriers and challenges spanned project management, logistical issues, integration of IT, and issues relevant to the identification of and recruitment of PCNs, general

**Table 2. Participant characteristics.**

|  | N | (%) |
|---|---|---|
| **Total** | 20 | (100) |
| **Gender** |  |  |
| Female | 11 | (55) |
| Male | 9 | (45) |
| **Employer** |  |  |
| NHS | 18 | (90) |
| Private sector | 2 | (10) |
| **Designation** |  |  |
| General Practitioner (GP) | 6 | (30) |
| Clinical Pharmacist (CP) | 3 | (15) |
| Practice Nurse Manager/lead, nursing associate (N) | 3 | (15) |
| Program/practice manager (PM) | 8 | (40) |

practices and patients. A list of the key emergent themes identified is presented in **Table 3** below. A detailed table with all supporting quotes is provided in **S3 Table**.

**Project management.** Some respondents expressed confusion and frustration due to shifting directives in the project, which they thought has led to a waste of time and resources:

> "*It started with being a project about working with UCLP. It then sort of got honed down to a project about hypertension. So a lot of the time, actually the funded time was lost doing other things before somebody decided this is what they wanted us to do*" [PM]

Further, the original BP@home national SOP, while useful, was also considered by some respondents as *"insufficient"* and still requiring substantial work and *"time to drill down into the logistics of how specifically this is going to work."* [PM]

Respondents also commented on the impact of the high rate of staff turnover, which required constant training of new recruits and contributed to *"the organisational memory loss"*. Finally, the BP@home program was perceived, especially by HCPs involved at the practice level, as being too *"top-down"* and not responsive enough in its approach:

> "*I feel that I've had very little ability to influence things at an ICS level. You know, we've had lots of frustrations with the delays and the BP machines and no matter how many times we've raised those concerns, nothing has really changed.* [GP]

This statement was further reinforced by one of the clinial pharmacists:

> "*I think I kind of echo what what was said at least in terms of influence of the [local PCN] level, I you know, we received some communications every now and then that was that was it*" [CP]

**Logistics.** A central logistical barrier identified by almost all respondents related to the reception, storing, distribution and tracking of BPMs, reinforced by the absence of a unified management template and leading to the potential waste of BPMs as a result.

In one of the ICSs, the delivery of 4,500 BPMs created *"a bit of a panic"* [PM]. The BP@Home initiative was described by this respondent as *"pretty chaotic"* with the distribution deemed as *"unprofessional"*. Another respondent reported a significant error with the

**Table 3. Thematic analysis of barriers and challenges to streamlined implementation of BP@home.**

| Category | Theme | Description |
|---|---|---|
| (1) Project Management | 1. Shifting directions & lack of national guidance | • Part of funded staff time lost due to changes in directives<br>• Fellows had to find alternative funding (inequity programs for example)<br>• SOP insufficient for immediate implementation |
|  | 2. Lack of resources (human, time, etc) | • Capacity in primary care is the main challenge<br>• Requires a person fully dedicated to the project |
|  | 3. Top-down approach & high staff turnover | • Limited influence of HCPs on processes<br>• Loss of organisational memory<br>• Requires constant re-training of new staff |
| (2) Logistical issues | 4. Storage & distribution of BPMs | • Complications & delays<br>• Unequal distribution to PCNs<br>• Waste of clinician's time<br>• Lack of XL cuffs |
|  | 5. Limited tracking capacity | • Very time consuming<br>• Practices do not have the required manpower |
|  | 6. Lack of awareness & unification of IT tools | • Lack of guidance on IT tools & systems to use<br>• Multiplicity of resources available can be overwhelming for GPs |
|  | 7. Limitations of existing tools (AccurX) | • AccurX requires a smartphone<br>• Links expire before patient start to enter readings<br>• Lack of automated feedback |
| (3) Engagement with PCNs & practices | 8. Differential capacities | • Some PCNs/practices not used to working together<br>• No financial incentive except BPMs<br>• Skill sets vary from practice to practice |
|  | 9. Substantial additional workload | Storing, tracking, management of readings might put off practices |
|  | 10. Lack of financial incentives | • No additional funding<br>• BPMs as only financial incentive |
| (4) Engagement with patients | 11. Limited & differential engagement | • Keeping patients engaged is a challenge & requires chasing<br>• Least engaged are those needing most care<br>• Limited human & IT resources to track & follow up |
|  | 12. Differential digital access & literacy | • Older and/or economically deprived patients might not have access to smartphone and/or internet |
|  | 13. Equity concerns | • Unequal distribution of BPMs might reinforce existing inequalities<br>• Issues with digital access & literacy might exclude at-risk patients, especially elderly and most deprived |

application for BP monitors which left their ICS without any monitors for the third wave of allocation. In addition, respondents reported an insufficient number of XL cuffs, which were sometimes limited to only one per practice. Accessing the BP monitors "*was complicated and time-consuming*" with some primary care sites still waiting for the machines while "*there's now like two office rooms at the federation that are just chock full of monitors.*" [PM]

For practices, especially the smallest ones, the lack of time and human resources, exacerbated during the Covid-19 pandemic, meant that they had to adapt the type and amount of information they could record and track since they "*do not have the manpower or infrastructure to keep an accurate log of [BPMs]*" [GP]:

> We initially requested the serial numbers [of BPMs] but then decided not to record them—it is very time-consuming and [we're] uncertain about the clinical relevance. It was a lot of work for a small practice. So we opted instead to record the cuff sizes. [PM]

Tracking and management issues were further reinforced by the absence of a unified template from NHSE to support the tracking of monitors and patients. Respondents also highlighted that most had little awareness of existing IT tools to help effectively track monitors and patients.

Finally, while respondents were mostly satisfied with AccurX which was the most commonly used tool to request and receive blood pressure readings from patients, some limitations were emphasised, including:

> the problem with AccurX is that is sounds as if it is text-based but it actually requires a smartphone (. . .) And we find that quite a lot of our AccurX requests fail either because the patient doesn't want to bother taking their BP for four days or because they just can't manage the tech. [PM]

One respondent complained about the links sent to patients using SMS requesting the readings expired automatically after a few days, and because this added a time constraint for patients, it may have further reduced patient compliance and the number of readings communicated to HCPs. Another respondent noted that the lack of automated feedback from AccurX may have limited the activation of patients using reminders, with the follow-up being reliant solely on the availability of HCPs.

**Recruitment and engagement of PCNs & practices.**   The lack of adequate, dedicated funding to support the recruitment and implementation of the BP@home program was perceived as an important barrier to the engagement of PCNs and practices:

> The only financial incentive for this is actually getting the monitors. There's no financial gain for a PCN, and that's why it's such a huge sort of issue. [PM]

This was particularly problematic due to the substantial additional workload associated with participating in BP@home. When approached, numerous practices saw this as: *"an extra piece of work"* and *"too complicated" [PM]*, especially in the context of the ensuing Covid-19 pandemic and national lockdown which severely disrupted all aspects of daily life.

> Broadly speaking, the initiative was highly welcomed. However, due to pressure from implementing vaccine rollout for Covid-19, resources were redeployed [PM]

This further emphasised the existing differential capacities of PCNs and practices. One respondent described PCNs as *"very variable beasts"* with some *"working very, very closely together"* where you can *"get a whole program out across the PCN"* while others described practices as *"forced to work together (. . .) but not actually really working together" [PM]*. Similarly, another HCP noted that *"skill sets vary from practice to practice" [PM]* thus affecting their capability to participate in this type of program.

Finally, these engagement issues led to equity concerns, in particular due to the unequal distribution between PCNs which might exclude or disenfranchise the most vulnerable patients:

> There are PCNs that have had lots and lots of monitors and their patients are benefiting, and you're having these populations who have high levels of deprivation, who again have missed out on these BP monitors. [PM]

**Recruitment and engagement of patients.**   Similar issues were reported regarding how patients were stratified or recruited into the BP@home pathway. While interviewees reported

a general ease of use of BPMs by patients, they found that many patients did not follow the recommendations:

> "Patients didn't need much help with using the BPM, they were quite comfortable using them. But many did not follow the request to send 4 readings a day, even after being sent a reminder and being provided a diary in the letter." [PM]

While the IT limitations with AccurX might have played an important role, respondents also highlighted other IT-related challenges that could explain some of the challenges encountered. Several HCPs reported some reluctance on the part of their patients due to their fear of losing face-to-face contact, especially in a context of increasing remote socialising, due in part to the Covid-19 lockdown but also to the already existing digitalisation of services. Another explanation for the limited engagement of patients was their immediate perception that their hypertension was not a priority in the context of the ensuing Covid pandemic where "*getting a jab" [PM]* and limiting physical contact were paramount. The prolonged duration of the BP@home initiative was also identified as a potential issue, along with the "experimental" nature of the program:

> It went on for too long and patients lost interest. It might have helped to have a timeline. If we don't dress it up as a trial, patients might take it more seriously. [PM citing feedback received from other care coordinators in the local PCN]

> Some of us [HCPs] actually believed this was a pilot—we were told this was a pilot- but it turns out general practice will now be doing this [BP@home] for ever [PM].

This lack of engagement was reported to be particularly acute among patients with multiple conditions and high levels of deprivation with whom engagement with primary care was already limited. There was also the impression that patients with high BP might feel discouraged by their readings and thus prefer not to measure, or communicate the readings, since those communicating their results the most or more frequently were the patients needing the least care as their hypertension was not impacted by multimorbidity. Another issue raised by a respondent was that once reassured that their BP is good, patients might not feel the need to keep self-monitoring and/or submitting readings.

## Facilitators and strategies

In response to the barriers and challenges identified in the previous section, respondents also described various strategies and facilitators to support the implemention of BP@home. While some were already being implemented in specific PCNs and practices, others were formulated as general recommendations to streamline not only the implementation of BP@home specifically but also of remote BP monitoring in general.

We classified strategies and facilitators into four main categories: (1) project management, (2) logistics (including technical and IT), (3) recruitment and engagement with PCNs and practices, and (4) recruitment and engagement with patients. Emergent themes are presented in **Table 4** below. A detailed table with supporting quotes is provided in **S4 Table**.

**Project management.**   One programme manager described that communication regarding BP@home was *"very good throughout the initiative"* and *"ongoing" [PM]*. The regular meetings with NHS England and at the ICS level were good opportunities to share information, discuss processes and systems and increase efficiency.

**Table 4. Thematic analysis of strategies and enablers to streamlining the delivery of BP@home.**

| Category | Theme | Description |
|---|---|---|
| **(1) Project management** | **Good communication with NHSE & within ICS** | • Regular meetings with NHSE & at ICS level<br>• Good opportunities to share information, discuss processes & systems and increase efficiency |
| | **Centralisation at practice & PCN levels** | • One person per practice with allocated time & resources to BP@home in charge of onboarding, follow-up, etc<br>• One person at PCN level guiding local teams (eg: NCL LTC clinical lead) |
| | **Task-sharing focused on LTCs** | • Task-sharing between different actors such as LTC clinical lead, lead pharmacists, reception representative & person who can advise and liaise feedback relative to their specific role |
| | **Incorporation into daily practice & reverse-thinking** | • Make it part of the day-to-day long-term condition reviews (not a separate project)<br>• Practices should anticipate and devise process & responsibilities for data/readings reception, follow-up with patients and tracking prior to entering the program |
| **(2) Logistics** | **BPMs on prescription / on loan** | • Would become a routine part of managing hypertension<br>• May improve the balance of who gets devices<br>• Would give patients more responsibility for the machine<br>• Weighing based on deprivation scores & lending of BPMs |
| | **Development of practice-specific processes & tools** | • System for managing results must be adapted locally<br>• Creation of local searches tools to track key metrics |
| | **Simplified templates & IT system** | • Streamline processes & resources, simplify template<br>• Identify a simple way of tracking BPMs & patients<br>• Use Proactive Care Framework for tracking recommendations for tracking metrics |
| | **Role of pharmacists** | • Well-placed to advise on logistics & liaising with patients (communication, double-check readings, motivation, etc) |
| **(3) Engagement of PCNs & practices** | **Additional funding** | • Already stretched system: additional work requires additional funding<br>Use alternative funding (e.g. equity funding) |
| | **Clinical targets as incentives** | • Tie participation in program as a way to achieve Qof & LTCs goals |
| | **Needs assessment & training** | • Needs assessment and training to make up for differential capacities between practices |
| **(4) Engagement of patients** | **NHSE onboarding material & integration of UCLP into EHR** | • Material praised as extremely clear and exhaustive, providing useful support for onboarding of patients<br>• UCLP integration provided initial identification and priorisation of eligible patients |
| | **Pragmatic/opportunistic onboarding** | • Contacting a limited number of patients (equal to the number of BPMs) and enrolling those who respond<br>• Matching of the BP@home priority list patients with those with pre-existing diabetes to focus on most at-risk patients |
| | **Person-centred care** | • Explain importance of BP tracking & support general self-management<br>• Customise frequency of contact by reducing requests of readings for those who have good results |
| | **Diversity of communication channels to fit each patient** | • Diverse communication channels required to communicate readings to avoid excluding patients, including non-digital options |

Other respondents emphasised the importance of task-sharing and coordination centred on long-term conditions (LTCs). Reflecting on the ICS one respondent was allied with:

> There is someone being appointed as the [ICS] LTC clinical lead and their job is basically going to be to go to practices and try and sort of establish what the problems are and what support they need, what resources they need, what's lacking in certain practices and PCN [PM]

Such centralised coordination at the ICS level was also recommended for practices that, according to several respondents, should all have a named person in charge of the BP@home, and that this local "champion" should be trained and funded:

> . . . because it takes time, and the practices that are successful with BP at home are the ones who have a person who has allocated time to be the responsible [GP]

Whereas one respondent previously countered this proposal, one participant recommended the incorporation of BP@home into routine general practice and that it should be considered business as usual:

It is here to stay. It is a priority as part of pushing proactive care. I do think it can become more and more sustainable, but we just need to keep tweaking and refining how we integrate it into our everyday approach to patients with long-term conditions and also how we support patients in their engagement, especially for groups who have been hard to engage [PM]

In order to support this integration into daily practice, processes should be based on NHSE recommendations but still be adapted locally so that they work in the real world even if they are tailored to be practice-specific:

Systems for managing results have to be developed for each practice as all practices work differently and it has to be decided who will respond to results etcetera. Some of these resources we have shared with practices existed on the NHS futures website and have been adapted for use locally [PM]

**Logistics.** A central, recurrent recommendation to the logistical challenges encountered in terms of storing, distribution and tracking of BPMs was to provide the devices on prescription. This was also presented as a way to deliver more equitable access to the machines. One respondent shared their ICS plan of recycling BPMs 3 to 4 times, after a few weeks with each patient. This 'recycling' was in line with the BP@home SOP to promote equity by avoiding a first-arrived, first-served logic which felt "irresponsible" and to limit waste and underuse. Respondents provided examples of how some PCNs intended to lend the monitors for a week to tier or priority group 3 patients and for 8/10 weeks and up to 6 months for tiers 1 & 2 patients. With a similar concern towards addressing inequalities, another respondent explained that the allocation of BPMs to practices was weighted based on deprivation scores and demographics.

Another central driver to the successful implementation of the BP@home initiative was the development of practice-specific processes since:

There are a lot of processes to be thought about and developed. And it still does for any practice because it happens at practice level: each practice needs to have systems for managing things they didn't use to manage, which is patients sending up results from home, and also onboarding them, giving equipment, etc. [PM]

In parallel, some respondents also emphasised the need for simplified, unified IT tools and tracking templates:

"*we've brought together quite a lot of resources, but actually it would be quite nice to have them much more streamlined and easily available. Maybe adapting the template and simplifying it. [PM]*

Another recommended strategy was to involve pharmacists in the planning of logistics due to their privileged relationship with patients as well as their experience in tackling logistical issues:

Pharmacists are brilliant at motivating patients as well as making sure that data is being recorded in the right way so that it's useful for them, for your IT people to pick up things on their searches and things like that [PM]

**Recruitment and engagement with PCNs and practices.** Several drivers to support the recruitment and engagement with PCNs and practices were identified. A central recommendation referred to increased incentives, including the use of clinical targets, such as QoF or local hypertension goals as part of broader LTC plans. In one ICS, a conditional financial incentive was provided:

[we] set up an incentive scheme as we recognised that money would be helpful. 130,000— so about 15,000 per borough for the eight boroughs, about two or three thousand per PCN, so very little per practice but it smoothed an awful lot of feathers and was very helpful. It also gave an opportunity to provide a plan: they had to put up a PCN team, (. . .) and confirm that ten unique patients had submitted readings remotely to receive the incentive [PM]

All respondents agreed that adequate funding was essential to ensure the sustained and efficient implementation of the BP@home programme since *"a lot of the work that's expected to be done is sort of out of an already stretched system that doesn't have actually any additional capacity". [GP]*

Another driver to support engagement with practices was the implementation of a needs assessment plan -–including training, role-playing, feedback and standardisation of materials -–to identify and address differential capacities within participating practices.

**Recruitment and engagement of patients.** One of the drivers supporting patient recruitment was the quality of the onboarding material provided by NHSE, praised by respondents as *"fantastic, extremely clear, a lot of support with IT, how to explain to patients that it would benefit them, and how"* thus leaving this respondent with *"no unanswered questions, we didn't feel like we were in the dark." [PM]*

In addition, the identification of eligible patients was facilitated by the integration of the UCLP priority list onto EMIS and SystmOne, *"which breaks patients down into different groups" [GP]*. However, this procedure usually yielded many more eligible patients than available monitors. Most practices described adopting a pragmatic or opportunistic approach to prioritising patients, such as those described below:

If we have only 20 machines, there is no point sending 200 texts: only send 20 texts saying "We identified you to benefit from this program". If the patient sends back "yes", call them back. (. . .) Pragmatically, we simply called back those who responded yes [GP]

In another practice, the team decided to consider the search for eligible hypertensive patients identified using the UCLP Proactive Care stratification tool who were known diabetics.

We looked at the priority list and then matched them with high-risk patients who have pre-existing diabetes. Then we sent en masse text messages to all the patients in the cohort to identify those who have a BPM at home. (. . .) then started to engage in terms of creating a management plan that is unique to our practice in that it included information about BP, the medication they were taking, along with instructions about how to monitor BP at home [CP]

The adoption of a more person-centred approach, which prioritises the patient's needs, understanding and acceptance of the initiative, was highly encouraged:

Patients have to understand why we're doing it and that is not just a thing the practice is asking for the benefit of the practice. They need to see that it's personally helpful for me to understand my blood pressure and to have this engagement and interaction with the clinician [PM]

The same respondent suggested one of the ways to better adapt to each patient's needs was to reduce the number of requests in case of satisfactory readings. In one practice for example:

If their BP was to target then we said OK, you don't need to do this for another 12 months and we automated it (. . .) because we don't want patient fatigue. [PM]

Finally, numerous respondents emphasised the necessity to maintain non-digital options for communicating results for patients with limited digital literacy and/or access in order to avoid excluding patients. Several practices reported using a multiplicity of channels as exemplified in the quotes below:

We use four different methods of sending in their BP readings: they can hand them at the reception by hand, they can email them to the practice email address, they can email the Excel spreadsheets that averages them for us [GP]

the remote monitoring requires the patient to at least have a mobile phone. So we have some older patients (. . .) or actually patients who maybe are from the slightly more deprived backgrounds who have mobile phones but don't have Internet access on their mobile phone. So that was OK because we sent the initial messages just as an SMS text message with the information and they could reply to the message if they wanted to bring their results in on paper. So we did have a cohort of patients who don't want to use Floreys* for whatever reason. They either got confused using a Florey* or they didn't have Internet access or their mobile phone was a more basic mobile phone without Internet access. So depending on the area, but it was usually not more than 10% of patients were still bringing paper copies in, but we still use the same protocol, the same SOP, to monitor them and so on.

**Future: An inevitable shift that requires adjusting.** Finally, it is important to note that remote monitoring of blood pressure was perceived by most respondents as inevitable, and even desirable, despite all its challenges:

It is definitely hard for GPs but in the long run, most would agree that it is a much more sound way of tracking people's BP due to the "white coat" syndrome [GP]

However, as noted by one respondent, this type of remote monitoring is still perceived by some HCPs as an alternative, additional option and not a replacement to face-to-face care and consultations:

It is going to be interesting to see what happens with community pharmacy in terms of task-shifting. Will it increase the number of people taking care of their health? This would be great though it does not replace a visit to the GP. It is not either or. Hypertension reviews will still be happening. [PM]

## Discussion

As remote BP monitoring is becoming more routine, it is crucial to investigate system and patient-level barriers to the successful implementation of remote monitoring of blood pressure in the home setting. To our knowledge, this is the first study that sought to characterise the enablers and barriers to the successful implementation of the NHSE-funded BP@home programme in the first two years after the initial launch in 2020 from the NHS perspective.

### Main findings

This study sought to explore the perspectives of programme managers and primary care workers across London about barriers and drivers for the successful implementation of the BP@Home initiative, and the future of remote blood pressure monitoring in general.

According to respondents at ICS/PCN level, their main challenges related to the differential engagement capacities of PCNs and practices, which raised equity concerns, a lack of financial incentives to support the substantial additional workload, unclear guidance from NHS England and finally the reception, distribution and storing of BPMs, especially without a unified IT tracking template. Regarding patient-facing HCPs, similar challenges were encountered in terms of the engagement capacities of patients and insufficient resources to support BP@home. In terms of logistics, respondents expressed some frustration with IT tools, as well as the necessity to manage a variety of communication channels to best adapt to each patient's needs and capacities.

Respondents also identified a multiplicity of strategies and drivers to support smoother adoption and implementation of BP@home. These included the use of clinical and financial incentives at the ICS/PCN level as well as increased task-sharing and training based on needs assessment. Another central recommendation was the provision of BPMs on prescription as well as engaging with pharmacists to support more efficient logistics. Finally, practice-based respondents also praised the quality of the onboarding material provided by NHSE as well as the integration of the UCLP criteria into the EHR. Combined with practice-specific, pragmatic approaches to onboarding patients. The provision of a more person-centred care approach was also recommended, along with the incorporation of BP@home into daily practice, especially with a named person, funded and trained in charge. Finally, the development of practice-specific processes and simplification of IT tools were also recommended.

### Comparison with existing literature

These findings align with existing literature suggesting that the barriers limiting the efficient and widespread implementation of remote self-monitoring among doctors and patients are cultural, structural, and financial [18, 19, 21].

Regarding HCPs, studies show that these barriers include the lack of adequate infrastructure and secure means of data transmission, which may prevent doctors from receiving patients' data and from interacting with them. There are also important costs relative to the implementation and maintenance of the IT tools as well as the training of HCPs [18]. HCPs interviewed in Scotland also expressed concerns about the additional workload and the responsibility to act immediately when faced with a continuous stream of readings [19]. Integration of the readings received through telemonitoring into routine data handling was also identified as crucial to the success of another initiative in Scotland [20]. Moreover, IT tools and systems should be more user-friendly, cost-effective, durable and with better safety standards [21]. The recommendations formulated by respondents regarding the future of the initiative were also in line with those of the American Heart Association and American Medical Association regarding the implementation of self-measured BP monitoring. These agencies

called for additional financial investment to support the education and training of individuals and providers, build health information technology capacity, and enhance reimbursement among other objectives [29].

While this research was focused on professionals' experiences and perspectives, respondents also commented on patients' potential barriers and drivers. These findings are also congruent with existing literature on the topic, highlighting the frequent inadequacy of remote monitoring programs to the needs of specific demographic groups such as the elderly, people with certain disabilities, who are not fluent in English or those lacking access to online technologies [4, 21]. Socio-economic inequalities also pose significant challenges to equal access, use, adoption and impact of eHealth services [22].

This research also has bearing on the emergent self-driven healthcare (SDH) movement in the UK [30], which seeks to empower individuals to self-care through a task-shifting approach based on leveraging the power of technology and self-generated data. The finding also offer insights into some of the structural and systemic factors which must be accounted for and addressed to efficiently improve health and wellbeing outcomes for all patients.

**Contextualisation of findings using the Consolidated Framework for Implementation Research (CFIR).**   Our study's findings, examined through the lens of the second iteration of the Consolidated Framework for Implementation Research (CFIR) published in 2022 [28], offer comprehensive insights into the implementation of the BP@home initiative in London's primary care setting.

*Intervention characteristics*. The BP@home initiative which is a complex intervention with multifaceted components primarily encountered challenges in its implementation due to resource constraints. These constraints align with the 'Intervention Complexity' and 'Resource Availability' constructs of the CFIR. The initiative's design, which required significant IT, human and financial resources, was also a critical barrier, reflecting CFIR's emphasis on both the intervention's adaptability and cost. However, the integration of eligibility criteria into electronic health records, which is a feature of the 'Design Quality and Packaging' construct, emerged as a facilitator. This enhanced the initiative's uptake but more work should be done at a national level to ensure that the distribution of blood pressure monitors does not exacerbate extant inequalities [31].

*Outer setting*. Our findings highlight the impact of 'Patient Needs and Resources' and 'External Policies and Incentives', whereas the pandemic's urgency necessitated rapid adaptation of healthcare practices, aligning with the 'Cosmopolitanism' construct. However, disparities in resource distribution, a component of the 'Outer Setting', exacerbated equity concerns, highlighting the need for policies that ensure equitable access to healthcare innovations.

*Inner setting*. The 'Inner Setting' of primary care practices significantly influenced the implementation. Resource inadequacy, aligning with the 'Readiness for Implementation' construct of CIFR, and the additional workload posed challenges. However, CFIR's 'Networks and Communications' construct was evident in the collaborative efforts among PCNs and practices, which facilitated smoother implementation. This collaboration, alongside CFIR's 'Culture' and 'Implementation Climate' constructs, will be pivotal in addressing logistical issues and promoting a supportive environment for the BP@home initiative.

*Characteristics of individual*. HCPs' perceptions and engagement played a crucial role as emphasized by CFIR's 'Knowledge and Beliefs about the Intervention' and 'Self-Efficacy' constructs. HCPs' pragmatic approach to patient onboarding and their adaptability reflect the 'Individual Stage of Change' and 'Personal Attributes', and highlight the importance of engaging and training HCPs to foster positive attitudes towards new health interventions.

*Process*. The 'Planning' and 'Engaging' constructs were also critical in success of the BP@home initiative. Effective project management and patient engagement strategies aligned

with these constructs, and both are recommended to overcome barriers. Additionally, the 'Executing' and 'Reflecting and Evaluating' constructs of CFIR suggest the need for continuous monitoring and adaptation of the initiative based on feedback from HCPs and patients. Our experience highlights that there was little time to reflecv and evaluate in real time during the roll-out of the BP@Home programme at a time of national crisis.

The CFIR framework provides a comprehensive lens to understand the complexities and challenges in implementing the BP@home initiative. Our study highlights the importance of considering all aspects of the CFIR framework–from the intervention's design to the external environment and individual characteristics–to ensure successful implementation. Future research should focus on exploring these dimensions in other regions and healthcare settings, contributing to a more nuanced understanding of implementing remote healthcare initiatives in the post-covid era.

## Implementation implications & authors' recommendations

Based on the participants' responses, the different barriers & challenges identified above, in particular the lack of XL cuffs, reception, storage & distribution issues, shifting directions and the limited resources, impacted the implementation of the BP@home program mainly through the delayed delivery of BPMs to patients as well as the limited capacity for keeping patients adequately engaged. However, the pre-existing drivers (including for example the good communication with NHSE) as well as the numerous strategies put in place by the HCPs involved in the implementation of the BP@home program in London, allowed for a good roll out of the initiative with most BPMs eventually reaching out to patients [32].

Drawing on the recommendations formulated by participants as well as published tliterature on the topic, we drafted key recommendations for each domain: project management, logistics, engagement of PCNs & practices and finally engagement of participants.

1. **Project Management Recommendations**

- Centralize project management at all levels (ICS, PCN, practice)

- Support task-sharing between specialty leads focused on LTCs

- Incorporate into BAU through supported training & capacity building

- Support reverse-thinking at practice level (plan logistics prior to entering BP@H) and support practice-specific processes

2. **Logistics Recommendations**

- Deliver BPMs on prescription

- Engage pharmacists in devising logistics & liaising with patients

- Simplify existing tracking templates

- Evaluate current IT tools/platforms to push for improvements while also exploring available IT alternatives

3. **Recommendations for effective engagement of PCNs & practices**

- Provide specific, adequate funding & training to HCPs based on needs assessment

- Use clinical targets (e.g., QoF) as incentives

  4. **Recommendations for effective engagement of patients**

- Provide person-centred care, adapted to each patient

- Maintain non-digital options for patients with limited digital access/literacy

- Future studies should explore patients' perspectives, experiences and needs, especially those from at-risk & underserved communities

## Limitations

Our study sample was small and from a localised area, London, so the findings of our study are not necessarily representative of the broader UK healthcare professionals involved in the BP@home programme. We also acknowledge that additional focus groups and interviews may have resulted in the identification of other emergent themes. Inevitably, the study sample included some selection bias [33], such that only those with an interest in, and or time availability for, sharing feedback on their involvement in the BP@home program responded. Also carrying focus groups remotely, while allowing greater flexibility and therefore easier participation for respondents may have limited access to non verbal cues and dynamics between participants. Finally, despite our effort to gather HCPs working at similar levels, power dynamics may still have been at play and affected the participation of some respondents in the discussion or the type of perspectives shared.

## Strengths

Despite its localised and relatively small sample of respondents, this study builds on the perspectives of a diversified group of healthcare professionals from four of the five London ICSs. These respondents were involved at all levels (ICS, PCN and general practice) as well as in different roles, including GPs, nurses, clinical leads and pharmacists. These respondents provided rich and useful insights into their personal and professional experiences delivering and supporting the BP@Home program. In addition, we feel that saturation of themes was achieved, as recommended in qualitative studies [34], since no new themes appeared in the last focus groups. Finally the choice of the Framework Methods aimed to keep as close as possible to the perspectives of respondents, without imposing an analytical framework a priori, except for the drivers versus barriers distinction. In addition, the inductive analytical categories (such as project managements, logistics, etc) identified by the authors are complemented in the discussion section by an association with key elements of the CFIR framework [28] to facilitate comparison with other studies.

## Conclusion

Programmes such as BP@home will become more common in primary care. To successfully support HCPs' aim to care for their hypertensive patients, their implementation must be accompanied by additional financial, human and training resources, as well as supported task-shifting for capacity building. Future studies should explore the perspectives of HCPs in other parts of the UK as well as patients' experiences, and personal drivers and barriers for routine monitoring of blood pressure in the contemporary setting.

## Supporting information

**S1 Fig. UCLP high blood pressure stratification and management framework.**
(DOCX)

**S2 Fig. BPM allocation recommended pathway.**
(DOCX)

**S1 Table. Interview guide.**
(DOCX)

**S2 Table. COREQ checklist.** Consolidated criteria for reporting qualitative studies (COREQ): 32-item checklist.
(DOCX)

**S3 Table. Thematic analysis of barriers & challenges identified by respondents (with supporting quotes).**
(DOCX)

**S4 Table. Thematic analysis of strategies & enablers identified by respondents (with supporting quotes).**
(DOCX)

## Acknowledgments

The authors thank NHS England Evaluation Cell London, all five London ICS leads and NHS staff for offering their time and support in helping us drive this qualitative research.

## Author Contributions

**Conceptualization:** Eva Riboli-Sasco, Austen El-Osta, Marie Line El Asmar, Manisha Karki, Gabriele Kerr, Ganesh Sathaymoorthy, Azeem Majeed.

**Data curation:** Eva Riboli-Sasco, Austen El-Osta, Gabriele Kerr.

**Formal analysis:** Eva Riboli-Sasco, Austen El-Osta, Gabriele Kerr.

**Funding acquisition:** Ganesh Sathaymoorthy, Azeem Majeed.

**Investigation:** Eva Riboli-Sasco, Austen El-Osta.

**Methodology:** Eva Riboli-Sasco, Austen El-Osta, Marie Line El Asmar, Manisha Karki, Gabriele Kerr.

**Project administration:** Austen El-Osta, Ganesh Sathaymoorthy.

**Supervision:** Austen El-Osta.

**Writing – original draft:** Eva Riboli-Sasco, Austen El-Osta.

**Writing – review & editing:** Marie Line El Asmar, Manisha Karki, Gabriele Kerr, Azeem Majeed.

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
