## [Decision Letter · Decision Letter 0]

7 Sep 2023

PONE-D-23-22751Investigating barriers & facilitators for the successful implementation of the BP@home initiative in London: primary care perspectivesPLOS ONE

Dear Dr. El-Osta,

Thank you for submitting your manuscript to PLOS ONE. After careful consideration, we feel that it has merit but does not fully meet PLOS ONE’s publication criteria as it currently stands. Therefore, we invite you to submit a revised version of the manuscript that addresses the points raised during the review process.

Please address the reviewers' concerns and feedback about your manuscript before re-submission and consideration for publication in our journal.

We look forward to receiving your revised manuscript.

Kind regards,

Lea Sacca

Academic Editor

PLOS ONE

4. We notice that your supplementary figures and tables are included in the manuscript file. Please remove them and upload them with the file type 'Supporting Information'. Please ensure that each Supporting Information file has a legend listed in the manuscript after the references list.

Reviewers' comments:

Reviewer's Responses to Questions

**Comments to the Author**

1. Is the manuscript technically sound, and do the data support the conclusions?

Reviewer #1: Partly

Reviewer #2: Partly

Reviewer #3: Yes

2. Has the statistical analysis been performed appropriately and rigorously? 

Reviewer #1: Yes

Reviewer #2: N/A

Reviewer #3: N/A

3. Have the authors made all data underlying the findings in their manuscript fully available?

Reviewer #1: Yes

Reviewer #2: Yes

Reviewer #3: No

4. Is the manuscript presented in an intelligible fashion and written in standard English?

Reviewer #1: Yes

Reviewer #2: Yes

Reviewer #3: Yes

5. Review Comments to the Author

Reviewer #1: Abstract: The abstract needs to be more concise.

Strengths and limitations: Please define separately.

Conclusion: Recommendations need to be addressed separately.

The article should be grammatically sound.

Reviewer #2: 1. The description of the data collection methods is not fully clear.

a. Were 5 focus groups and 1 one-on-one semi-structured interview conducted? If focus groups were used, authors should describe key characteristics of the focus groups, per generally accepted guidelines. For example, how many people attended each focus group; where focus group attendees homogenous in terms of occupation to facilitate frank discussions? Who facilitated the focus group discussions? I suggest the term “interviews” be used only for interviews and not for focus groups, as focus groups constitute a method different from interviews.

b. Quotes in the result section and S5 and S6 Appendices should specify the respondents’ role, to the extent doing so does not compromise anonymity.

c. Please cite your methods.

d. Once methods are clarified, strengths and limitations should reflect these methods.

2. I think the results and discussion sections would be enhanced by a conceptual framework, e.g., an implementation framework. For example, do logics/project management factors map to an “early” stage of implementation whereas patient onboarding stage factors to a “later” stage? Furthermore, it is not clear what is meant by “system level” per the paragraph about the aim of the study? Does “system level” denote NHS level? Or do the authors mean different system levels, e.g., ICS level, PCN level, GP level? Which barrier/facilitator is “system” level? Which barrier/facilitator is at the ICS level versus GP level, if such a distinction is reasonable? The authors should be clear about what is meant by “system level”, and a conceptual framework may help with this clarification.

3. I think the authors should also describe implementation implications based on the combined findings of this study and recent studies (references 15-22).

4. Minor issue: These are some words that are difficult for an American reader to understand, e.g., “crib sheet” (Interview guide?), and from S6 Appendix, “Flory” and “flurry”.

Reviewer #3: First and foremost, I commend the authors for this insightful study that delves into the barriers and facilitators for the successful implementation of the BP@home initiative. The manuscript is coherently written and follows an organized format. Furthermore, the adherence to COREQ guidelines is a commendable aspect of your methodology. I offer the following comments for your consideration:

Methods

2. It's mentioned that a total of 20 individuals took part in this study. Could you elaborate on how this sample size was determined to be sufficient?

The manuscript states that the sample size of 20 was felt adequate to achieve saturation. However, it is not explicitly clarified whether data saturation was indeed reached.

Can you specify how many individuals were approached for the interview? It would be beneficial to know if everyone agreed to participate or if some declined or opted out during the process.

Regarding the relationship with participants: It would enhance the rigor of your study if you can clarify any pre-existing relationships between the data collectors and the participants to ensure transparency about potential biases. Were the interviewers known to the participants, or were they neutral entities?

Findings

Under the "Barriers" section, the discourse related to the theme of "project management" suggests that the study primarily focused on evaluating the BP@home program. There seems to be a slight divergence from the initial goal, which was understanding impediments to its implementation. For instance:

How do the shifting direction and the reduced timeframe for the project's roll-out impact the program?

Were there any repercussions due to delays in the project's implementation? It would enrich the findings if you could provide insights into how such challenges were managed or addressed.

6. PLOS authors have the option to publish the peer review history of their article (what does this mean?). If published, this will include your full peer review and any attached files.

Reviewer #1: No

Reviewer #2: No

Reviewer #3: No

---

## [Author Response · Author response to Decision Letter 0]

14 Dec 2023

Dear editor & reviewers,

Thank you very much for your thorough & valuable review of this paper. Please see below our responses to each of your comments.

Editor

1. Thank you. We have revised the manuscript following PLOS ONE’s style requirements, including those for file naming. We also removed funding or competing interests information at the end of the manuscript and only kept the Acknowledgments section.

2. Thank you for raising this. After checking the following guidance provided on PLOS website: “For studies analyzing data collected as part of qualitative research, authors should make excerpts of the transcripts relevant to the study available in an appropriate data repository, within the paper, or upon request if they cannot be shared publicly”, we believe that Supplementary files S5 and S6 do contain the minimal anonymized data set necessary to replicate our study findings. In fact, these include all supporting quotes for each identified theme.

We have revised our cover letter to reflect this by adding the following sentence: “The minimal anonymized data set necessary to replicate our study findings have been provided as Supporting Information files.”

3. Thank you. We have removed the “Ethics” section at the end of the paper (after the Conclusion).

4. Thank you. We have removed our supplementary figures and tables from the manuscript file and saved them as individual Supporting information files.

Reviewer #1

Thank you for all your comments. We have made the following revisions:

- We have revised the abstract to make it more concise.

- We present the strengths and limitations in two separate sub-sections of the discussion.

- Authors’ recommendations are addressed separately, along with the implementation implications.

- We have further revised the paper to ensure it is grammatically sound. 

Reviewer #2

Thank you for all your comments. We have made the following revisions:

1a. We have provided additional details regarding the focus groups, including who lead them and who were the participants, both in the text (“All focus groups were facilitated by AEO”) and by the addition of “Table 1: Composition of Focus Groups”. 

We also revised the wording across the manuscript to use “focus groups and interviews” rather than just “interviews” where appropriate.

1b. All quotes in the manuscript and supplementary files specify the respondent’s role. We had initially thought of numbering them (e.g. GP 1, PM 4, etc) but thought this might compromise anonymity. Therefore, only the role has been specified (as suggested), considering that this is sufficient to “situate” each quote’s perspective without compromising anonymity.

1c. We have specified the references supporting our methodological choices, including the data collection tools, sampling methods, and the analytical framework supporting data analysis.

1d. We have added some of the limitations and strengths related to the methodological choices we made.

2. We used CFIR to contextualise our findings, by mapping them to key CFIR constructs, and to help guide the development of congruent recommendation’s to streamline the delivery of BP@home initiative 

3. We added a section in the Discussion entitled “Implementation implications & authors’ recommendations” where we describe the implications of our findings, combined with recent studies, on the implementation of BP@H. 

4. We have replaced “crib sheet” by “interview guide” across the manuscript and supplementary information. We also added an asterisk and explanation of what a Florey is: * Florey is a feature in Accurx designed for collecting structured data and screen/monitor patients remotely. See this link for more details https://support.accurx.com/en/articles/3542649-florey-what-is-florey

Reviewer #3

Thank you for all your comments. We have made the following revisions:

1. Thank you for your supportive statement which is much appreciated.

2 (Methods) We have added a sentence regarding this in the Data Collection section: “Of those who responded and agreed to take part, two eventually could not attend any of the focus groups scheduled due to time constraints. However no additional interviews or focus groups were organized as theme saturation had already been achieved.”

We also rephrased our sentence in the Strengths section as follows: “In addition, we feel that saturation of themes was achieved, as recommended in qualitative studies [47], since no new themes appeared in the last focus groups.”

In addition we have clarified the number of people approached and whom declined in the Data Collection section.

Finally, regarding the relationship with participants, we have specified that “3 participants were previously known to study authors by virtue of participating in a local clinical research group monthly online meetings.”

3 (Findings) We added a section “implementation implications & authors’ recommendations” which aims to clarify these aspects. We have also contextualised our findings using the CFIR frameworks, mapping them to key CFIR constructs.

---

## [Decision Letter · Decision Letter 1]

1 Feb 2024

Investigating barriers & facilitators for the successful implementation of the BP@home initiative in London: primary care perspectives

PONE-D-23-22751R1

Dear Dr. El-Osta,

We’re pleased to inform you that your manuscript has been judged scientifically suitable for publication and will be formally accepted for publication once it meets all outstanding technical requirements.

Kind regards,

Lea Sacca

Academic Editor

PLOS ONE
---

## [Editor Report · Acceptance letter]

21 Feb 2024

PONE-D-23-22751R1 

PLOS ONE

Dear Dr. El-Osta, 

I'm pleased to inform you that your manuscript has been deemed suitable for publication in PLOS ONE. Congratulations! Your manuscript is now being handed over to our production team.

Kind regards, 

on behalf of

Dr. Lea Sacca 

Academic Editor

PLOS ONE